# The Molecular Landscape of Thymic Epithelial Tumors: A Comprehensive Review

**DOI:** 10.3390/ijms25031554

**Published:** 2024-01-26

**Authors:** Lisa Elm, Georgia Levidou

**Affiliations:** Department of Pathology, Nuremberg Clinic, Paracelsus Medical University, 90419 Nuremberg, Germany; lisa.elm@klinikum-nuernberg.de

**Keywords:** thymic epithelial tumors, TETs, thymoma, genetic alterations, signaling pathway, NGS, molecular diagnostic

## Abstract

Thymic epithelial tumors (TETs) are characterized by their extreme rarity and variable clinical presentation, with the inadequacy of the use of histological classification alone to distinguish biologically indolent from aggressive cases. The utilization of Next Generation Sequencing (NGS) to unravel the intricate genetic landscape of TETs could offer us a comprehensive understanding that is crucial for precise diagnoses, prognoses, and potential therapeutic strategies. Despite the low tumor mutational burden of TETS, NGS allows for exploration of specific genetic signatures contributing to TET onset and progression. Thymomas exhibit a limited mutational load, with prevalent *GTF2I* and *HRAS* mutations. On the other hand, thymic carcinomas (TCs) exhibit an elevated mutational burden, marked by frequent mutations in *TP53* and genes associated with epigenetic regulation. Moreover, signaling pathway analyses highlight dysregulation in crucial cellular functions and pathways. Targeted therapies, and ongoing clinical trials show promising results, addressing challenges rooted in the scarcity of actionable mutations and limited genomic understanding. International collaborations and data-sharing initiatives are crucial for breakthroughs in TETs research.

## 1. Introduction

Thymic epithelial tumors (TETs) originate from the thymic epithelial cells, which constitute the lymphoid organ situated in the anterior mediastinum known as the thymus [1]. Comprising predominantly epithelial cells and lymphocytes, the thymus serves as a site where precursor cells migrate and undergo differentiation into lymphocytes. Subsequently, a significant proportion of these lymphocytes undergo destruction, while the remaining cells migrate to various tissues, where they differentiate into T cells [1].

The aberrant proliferation of epithelial cells results in the development of TETs, namely thymoma and thymic carcnoma (TC) [2]. TETs are rare neoplasms, accounting for approximately 0.2% to 1.5% of all human malignancies [3]. Despite their low frequency among adults, TETs stand out as the most frequently occurring tumors in the anterior mediastinum [3]. These tumors, especially thymomas, possess distinctive biological characteristics and are linked to autoimmune paraneoplastic diseases like myasthenia gravis (MG) [3].

The histological classification of TETs is made according to the World Health Organization (WHO) criteria, which rely on the morphology of cancer cells, the degree of atypia, and the quantity of intratumoral thymocytes [4,5]. Histological classification alone is, however, insufficient to distinguish biologically indolent from aggressive TETs. Treatment outcomes and recurrence results appear to have a stronger correlation with the invasive and infiltrative characteristics of tumor cells [3]. Consequently, histological benign thymomas may exhibit aggressive behavior [3]. Survival rates vary, at 12.5% for invasive and 47% for non-invasive TETs over a 15-year period [1]. Among the several histological subtypes of TETs, TC is identified as the most aggressive variant, displaying an increased tendency towards metastatic dissemination [4,5]. Given the challenges in differentiating TETs among cases with potentially benign or malignant behavior, researchers try to find more precise molecular tools for this purpose. In this context, exploring mutations and signaling pathways becomes pivotal in enhancing the overall understanding of tumor biology in TETs.

The present review aims to provide a methodically grounded overview of the genetic-alteration and -pathway activation in TETs, with a focus on the integration of modern molecular techniques, such as Next Generation Sequencing (NGS)-based investigations. In recent years, genomics, particularly with the advent of these technologies, has marked a revolutionary advance in the molecular characterization of tumors. This technological breakthrough has empowered researchers to transcend traditional boundaries and try to scrutinize the genetic foundations of TETs, aiming for the recognition of parameters not only for diagnostic and prognostic purposes but also with potential therapeutic implications.

## 2. Genetic Alterations in Thymic Epithelial Tumors

### 2.1. Next Generation Sequencing

Numerous NGS investigations have recently tried to comprehensively explore the genetic foundation of TETs [6,7,8,9,10,11,12,13,14,15,16,17,18,19,20,21,22,23]. Due to the rarity of TETs, only limited studies have collected high-throughput sequencing datasets. Nonetheless, advancements in various molecular tests, especially NGS methods, have unveiled that TETs demonstrate the least tumor-mutation burden among adult cancers [6]. Several NGS studies emphasize this low mutation burden in TETs. Yamaguchi et al., for example, did not find any mutations in 19 of 24 investigated cases, with only three cases demonstrating *KRAS* and non-synonymous *HRAS* mutations and two having low *DNMT3A* mutations [7].

The key tumor-suppressor genes, *P53* (*TP53*) and *EGFR*, commonly mutated in human cancers, have also been scrutinized in thymomas. A recent study by Syahruddin et al. identified *EGFR* Exon 18 mutation (E709K) and a nonsense mutation in a small subset of 7.4% of cases [8]. *TP53* Exon 6 mutations, including both missense and nonsense mutations, were also detected only in 7.4% of cases. Both results suggest a limited role of *EGFR* and *TP53* in the pathogenesis of thymomas [8]. *TP53* mutations, however, seem to be prominent in TCs, especially in their highly aggressive forms [9]. *TP53*, often co-mutated with *BCOR*, dominates mutations in TCs, while *ASXL1* (p.E657fs) and *DNMT3A* (p.G728D) mutations were observed in Type B3 thymomas [10]. NGS, in particular Whole Exome Sequencing (WES) and Targeted Sequencing (TS), in TETs has revealed a distinctive missense mutation in *GTF2I*, which is a general transcription factor [11]. This mutation, particularly p.Leu404His, has been reported to be specific for TETs, mainly in Type A and AB thymomas, with a presence in 76–83% of cases, and which occurs less frequently in other subtypes, notably only in 8% of TCs [12]. Hsieh et al. also report the presence of another point *GTF2I* mutation, namely p.Leu424His, in multinodular thymoma with lymphoid stroma (MN-T) and suggest that this mutation consistently characterizes this histological subtype [13]. However, there are regional variations; studies in Japanese patients identified *GTF2I* mutations in all thymoma types except for Type B3, which may, in a few cases, show *SMARCB1* and *STK11* gene mutations [11]. In these studies, *GTF2I* mutation was accompanied with *HRAS* and *NRAS* mutations, suggesting a potential exclusivity to indolent thymomas [6,11]. Further studies, including a recently meta-analysis, confirmed that *GTF2I* mutations, along with *TP53* and *HRAS*, are prevalent in thymomas, contributing to disease onset and progression [14]. Furthermore, the persistence of specific *GTF2I* mutations in certain subtypes could suggest a strong genetic link between them, which is exemplified in MN-T and Type A/AB thymomas [13,14].

Apart from *GTF2I*, a very small subset of genes exhibits recurrent mutations in TETs, occurring in at least 3% of the cases [12]. These genes include—in addition to *HRAS* and *TP53*—*CYLD*, *PCLO*, and *HDAC4* [12]. *HRAS* mutations have been reported to be prevalent in Type A and AB thymomas, being the second most mutated gene in these tumors with a frequency of 7% [12]. *PCLO* and *HDAC4* mutations occurred in various thymoma subtypes and TCs, each with a frequency of 3% [12]. In summary, commonalities and differences in the identified genes of various WHO subtypes of TETs have been observed [12]. A detailed description of these alteration among several TETs is presented in Table 1. As shown in this table, common genes found in multiple subtypes include GTF2I, *HRAS*, *TP53, HDAC4*, *PTPRB*, and *NOD1*. Types A, AB, and B2 thymomas share many of these genes, with additional occurrences of *PAX7* and *CSF1R* in Type A thymoma and *ZMYM3* in Type AB thymoma. Individual genes specific to certain subtypes include *BCOR* in Type A thymoma, *PBRM1* in Type AB thymoma, and *BRD4* in Type B2 thymoma [12]. TCs exhibit an expanded genetic diversity, incorporating many genes which are also present in thymomas. However, TCs additionally feature *CYLD*, *FGF3*, *BRD4*, and several recurring somatic mutations such as *TET2*, *SETD2*, *FBXW7*, and *RB1* [12]. The absence of a *GTF2I* mutation in TCs, as noted by Saito et al., highlights a specific genetic distinction [15].

Szpechciński et al. focused on 15 genes, and found pathogenic somatic single-nucleotid variants (SNVs) in *TP53*, *ERBB2*, *KIT*, and *KRAS* in 29.4% of TCs, while thymomas did not display any confirmed pathogenic SNVs. Rare SNVs in *ERBB2*, *KIT*, and *FOXL2* were identified in 16% of thymomas, with common germline SNVs in *TP53*, *ERBB2*, and *KIT* [16]. Additionally, the authors documented *AKT3*, *ALK*, *CSF1R*, *FGFR4*, *KRAS*, *NRAS*, and *PIK3CA* as common mutated genes in thymomas [16]. In the same context, Petrini et al. detected in TCs repetitive mutations in widely recognized cancer-associated genes, excluding *TP53* and *CYLD*, in *CDKN2A*, *BAP1*, and *PBRM1* [17]. Additionally, Okuda et al. recognized TC mutations in six tyrosine kinase genes *KIT*, *DDR2*, *PDGFRA*, *ROS1*, and *IGF1R* [18]. Similary, Enkner et al. found mutations in *ALK*, *ATM*, *CDKN2A*, *ERBB4*, *FGFR3*, and *NRAS* in 16 of 35 cases [9]. Regarding *KIT* locus, another study reveals that the genome copy number gains occurs in 6–12% of cases as well as at *AHNAK2* locus [15]. An NGS examination by Alberobello et al. revealed mutations within the *PIK3R2* gene in TCs, responsible for encoding a regulatory subunit of PI3K [19]. A subsequent analysis detected distinct mutations in various PI3K subunit genes in an alternative cell line and numerous primary TC samples [19]. These mutations encompassed two catalytic subunits (*PIK3CA* and *PIK3CG*) as well as an additional regulatory subunit (*PIK3R4*) [19]. Yang et al. found in a WES study that TCs showed a higher frequency of alterations in *MYO16* (33% vs. 3%) and a lower incidence of *ZNF729* changes (0% vs. 35%) in comparison to thymomas [20]. Moreover, they identified a number of significantly mutated driver genes, which include *PABPC1*, *ZNF208*, *PAK2*, *ZNF626*, *GTF2I*, *KMT2D*, *NOTCH1*, and *RB1* [20]. In a WES study conducted by Yang et al., *ZNF429* emerged as the gene displaying the highest frequency of mutations in TCs, accounting for 36% [21]. Additionally, exclusive mutations in TCs were identified in *BAP1* (14%), *ABI1* (7%), *BCL9L* (7%), and *CHEK2* (7%), while *ZNF721* mutations (14%) and *PABPC1* (14%) were uniquely associated with thymomas [21]. Common somatic mutations observed in the TC subgroup, compared to the control group, included *ZNF429*, *ZNF208*, *BAP1*, *ERBB4*, *GNAQ*, *ABI1*, and *BCL9L* [21].

Another interesting finding is the observation of *POLE* mutations in two of the five investigated cases of an exceedingly rare thymoma subtype, namely metaplastic thymoma (MT), which histologically sometimes resembles Type A thymoma [22].

Various investigations have also concentrated on examining samples of tumor tissue from individuals who have previously received chemotherapy [23]. The analysis of pre-treated TET patients reveals somatic mutations in various genes, contributing to our understanding of the genetic landscape associated with disease progression. In this context, Wang et al. examined 197 cancer-associated genes in pre-treated TET patients, detecting somatic mutations in 39 genes [23]. The prevalence was 62% in TCs and 13% in thymomas, with recurrent mutations in *BAP1*, *BRCA2*, *CDKN2A*, *CYLD*, *DNMT3A*, *HRAS*, *KIT*, *SETD2*, *SMARCA4*, *TET2*, and *TP53* [23].

Furthermore, Yang et al. conducted an analysis for a comprehensive overview of the genetic-alteration landscape of somatic mutations in “The Cancer Genome Atlas thymoma” (TCGA-THYM) database and compared the results with their WGS study. In the TCGA-THYM database, the most common somatic mutations in thymomas also included the genes *GFT2I* (50%) and *HRAS* (8%), as well as the genes *TTN* and *MUC16*, each with 7% (Figure 1) [21].

### 2.2. RNA Sequencing

Wang et al. utilized Ribonucleic Acid (RNA) Sequencing (RNA-Seq) and Fluorescence In Situ Hybridization (FISH) to explore gene fusions in thymomas. In 80% of MT cases, Wang et al. identified a *YAP1*–*MAML2* rearrangement, which was not present in Type A thymomas [22,24] and which could help the differential diagnoses between these entities. Figure 2 illustrates the composition of the dual break-apart probes used by Wang et al. in the FISH analysis, along with positive FISH samples demonstrating the *YAP1*–*MAML2* gene fusion in thymomas.

A similar observation from Vivero et al. [25] suggested that MTs can be distinguished by the presence of *YAP1*–*MAML2* fusions and the absence of *GTF2I* mutations in contrast to Type A and AB thymomas; therefore, they represent a distinct and clinically benign entity [24]. Moreover, MTs reportedly exhibit a chromosome 11 inversion [22,24].

Ji et al. conducted a further investigation on fusion genes in thymomas, identifying 21 fusions in 25 patients [26]. Among them, *KMT2A*–*MAML2*, *HADHB*–*REEP1*, *COQ3*–*CGA*, *MCM4*–*SNTB1*, and *IFT140*–*ACTN4* exhibited a significant expression, which was unique to thymoma samples [26]. Furthermore, Type B2 and B3 thymomas were rarely associated with *KMT2A*–*MAML2* translocations involving different combinations of exons 8, 9, 10, or 11 of *KMT2A*, and exon 2 of *MAML2* to varying extents [27]. Additional fusion transcripts documented in thymomas include *FABP2*–*C4orf3* and *CTBS*–*GNG5* [26].

The study of Ji et al. also revealed abnormal long non-coding RNAs (lncRNAs) and microRNAs (miRNAs) in the regulatory network [26]. The authors highlighted 65 distinct expression patterns of lncRNAs in thymomas, such as AFAP1–AS1, LINC00324, ADAMTS9–AS1, VLDLR–AS1, LINC00968, and NEAT1 and 1695 overexpressed miRNAs, validating their expression patterns through confirmation with “The Cancer Genome Atlas” (TCGA) database [26]. Moreover, elevated expression of AFAP1–AS1 and reduced expression of LINC00324 significantly influenced patients’ survival outcomes [26].

Abnormal miRNA expression influences immune cell functions and tumor progression. Wang et al. identified miR-130b-5p, miR-1307-3p, and miR-425-5p as predictive parameters for TET patients [28]. Enkner et al. found significant differences in the miRNA repertoire between Type A thymomas and TCs, with contrasting expression of C19MC and C14MC miRNA clusters [9]. In concordance, Radovich et al. reported that the sizable miRNA C19MC cluster, located on chr19q13.42, is a notable distinguishing factor among various thymoma subtypes [29]. This cluster exhibited significant overexpression in Type A and AB thymomas, while its expression was virtually negligible in other thymomas and non-tumor tissues (Figure 3) [29].

TCs exhibit also an altered expression in non-clustered miRNAs, including upregulation of miR-21, miR-9-3, and miR-375, and reduced expression of miR-34b, miR-34c, miR-130a, and miR-195 [9].

Most studies on tumorigenesis have focused on examining genetic alterations at the level of individual genes, with limited investigations in this area in TETs. Yu et al. examined 31 thymoma samples and identified 292 genes with overexpression of more than twofold, including six previously identified pivotal oncogenes (*FANCI*, *NCAPD3*, *NCAPG*, *OXCT1*, *EPHA1*, and *MCM2*) [30]. Among the highly upregulated genes were *CCL25*, *HIST1H1B*, *SH2D1A*, *DNTT*, *PASK*, *CENPF*, *HIST1H2BD*, *S100A14*, and *NPTX1* [30]. Additionally, 596 genes were downregulated, with *PLIN1*, *MYOC*, *ADH1A*, *FABP4*, *ADIPOQ*, *ADH1C*, *MGST1*, *LPL*, and *CIDEC* among the top 10 of them [30]. Chromosomal regions with notable aberrations in thymomas were found on chromosomes 1, 2, 6, 10, 15, and 19, with chromosomes 1 and 19 harboring the most of them. Interestingly, chromosome Y did not show any chromosomal aberrations [30].

Moreover, distinct expression profiles have been observed among TET subtypes, and are associated with development, immunity, and cancer [31]. For instance, Wu et al. found elevated expression of the four specific circular RNAs (circRNAs) hsa_circ_0001173, hsa_circ_0007291, hsa_circ_0003550, and hsa_circ_0001947 in TET patients, correlating with immunological imbalance [32].

### 2.3. Quantitative Real-Time PCR

Quantitative Real-Time PCR (qRT-) PCR has also been employed to confirm alterations in gene expression in TETs. Yu et al. in a messenger RNA (mRNA) study demonstrated increased expression of *E2F2*, *EPHA1*, *CCL25*, and *MCM2* as well as decreased expression of *MYOC*, *FABP4*, *IL6*, and *CD36* [30]. Interestingly, *EPHA1* showed a significant mRNA overexpression in 71.0%, and *MCM2* in 61.3% of the cases [30].

Vodička et al. utilized reverse-transcription–quantitative PCR (RT-qPCR) to analyze the mRNA expression levels of *CTNNB1*, *CCND1*, *MYC*, *AXIN2*, and *CDH1* [33], which were overexpressed in thymomas compared to control samples, with the exception of *AXIN2* in Type B thymomas [33]. In this case, mRNA expression exhibited a gradual increase from Type B1 to Type B3 thymoma [33]. Notably, thymomas Type A showed an exclusive significant increase in *AXIN2* mRNA expression [33].

Xi et al. explored the differential gene expression in thymoma-associated myasthenia gravis (TAMG) and non-myasthenia gravis thymoma (NMG), revealing 169 genes with distinct expression levels; 6 of them were overexpressed in T cells, namely *ATM*, *SFTPB*, *ANKRD55*, *BTLA*, *CCR7*, *TNFRSF25* [34]. Additionally, in the comparison of the transcriptomes between TAMG and NMG, Yu et al. observed an upregulation of 5 genes (*PNISR*, *CCL25*, *NBPF14*, *PIK3IP1*, and *RTCA*) by more than twofold, while over 30 genes were downregulated by more than a twofold decrease in TAMG [30]. *GADD45B*, *SERTAD1*, *TNFSF12*, *MYC*, and *ADPRHL1* were the most downregulated; these results were confirmed using qRT-PCR [30]. Another interesting study found a reduced expression of miR-20b in TAMG [35], which was identified to target *NFAT5* and *CAMTA1*. The presence of miR-20b in cultured cells led to the suppression of *NFAT5* and *CAMTA1* expression, whereas an inverse correlation between miR-20b and *NFAT5*/*CAMTA1* expression levels in patients with TAMG have been observed [35].

Several studies have tried to link gene-expression levels with tumor aggressiveness. For example, Gökmen-Polar et al. report a nine-gene signature predicting metastatic potential in thymomas, with genes like *AKR1B10*, *JPH1*, and *NGB* associated with increased metastasis [36]. Conversely, genes demonstrating decreased activity in thymomas prone to high metastasis encompass *DACT3*, *SLC9A2*, *PDGFRL*, *FCGBP*, *PRRX1*, and *SERPINF1* [36]. Yukiue et al. observed a significant increase in the mRNA expression of *TIMP-1* in stage II–IV compared to stage I thymomas [37]. Similarly, *MMP-1* mRNA expression has been reported to be elevated in invasive compared to noninvasive thymomas [37].

In a recent study conducted by Radovich et al., consensus clustering methods were utilized to categorize thymoma samples, and the created Reverse Phase Protein Array (RPPA) clusters showed a notable correlation with the WHO histological subtype [38]. Both, single-platform and multi-platform PARADIGM analyses were employed, integrating copy number and RNA expression data to discern genomic features superimposed onto the TumorMap for cluster differentiation [38]. The PARADIGM results indicated an increase in tumor-suppression genes, particularly *p53*, and a decrease in oncogenes such as *MYC*/*Max*, *MYB*, and *FOXM1* in the A-like cluster [38]. Conversely, the AB-, B-, and C-like clusters displayed a downregulation of tumor-suppression genes (*p53* and *TAp73a*) and an upregulation of oncogenes (*MYC*/*Max*, *MYB*, *FOXM1*, and *E2F1*) [38]. This observation aligns with the extensively documented increased clinical aggressiveness observed in Type B thymomas and TCs [38].

### 2.4. Chromatin Immunoprecipitation Sequencing

Moreover, an additional inventive strategy entails utilizing Chromatin Immunoprecipitation Sequencing (ChIP-seq) to discern distinct interactions between proteins and Deoxyribonucleic Acid (DNA). Chromatin-associated proteins are crucial in coordinating the spatial and temporal control of gene expression [39]. It is imperative to ascertain the genomic locations where these proteins establish binding, as they are integral to unraveling gene regulation intricacies and facilitate the exploration of pathways pertinent to tumor progression [39]. ChIP-seq precisely pinpoints genomic regions to which specific proteins such as transcription factors attach, thereby playing a pivotal role in comprehending the regulation of gene expression and the associated chromatin alterations [39].

Yuan et al. used immunohistochemistry to investigate *SOX9*, as a potential oncogene and therapeutic target in various cancers [40]. To determine differentially expressed genes (DEGs) regulated by the transcription factor *SOX9*, potential target genes were retrieved from the “ChIP Enrichment Analysis” (CHEA) databases. These databases are specifically tailored for the identification of target genes based on the published studies involving ChIP-chip, ChIP-seq, and other profiling methods for transcription factor binding sites [40]. While its significance in TETs was uncertain, the study identified *SOX9* as a potential diagnostic and prognostic marker [40].

### 2.5. Methylation Analysis

The methylation analysis facilitates an in-depth examination of epigenetic changes, providing insights into the regulation of gene expression [41]. Prompting alterations in gene expression, epigenetic changes involve adjustments in DNA methylation, post-translational modifications on histone tails, and disrupted expression of non-coding RNA molecules (ncRNAs) [41]. These modifications are progressively acknowledged as factors in tumorigenesis, contributing to genome instability, chromosome abnormalities, activation of transposable elements, elevated expression of proto-oncogenes, and inhibition of tumor suppressor genes [41]. There is indeed a growing interest in identifying epigenetic biomarkers for cancer, with potential applications in clinical settings for diagnostic or prognostic purposes, or as innovative targets for therapeutic interventions [41].

Many studies already investigated the methylation status of TETs. Bioinformatic analysis of the TCGA dataset identified 5155 hypermethylated and 6967 hypomethylated Cytosine-phosphate-Guanines (CpG) sites in Type A–B3 thymomas and TCs, respectively, with 3600 located within gene promoter regions. A total of 144 genes were subject to silencing due to promoter hypermethylation, while 174 mRNAs exhibited upregulation [42]. Cox regression analysis demonstrated a significant association between the methylation levels of 187 sites and overall survival in TET patients, where cg05784862 (*KSR1*), cg07154254 (*ELF3*), cg02543462 (*ILRN*), and cg06288355 (*RAG1*) emerged as independent prognostic factors [42].

TCs with *TET2* mutations seem to show more hypermethylated genes, correlating with downregulated gene expression. In a recent study, nine out of thirty nine mutated genes (23%) were involved in epigenetic regulation, with 34% of TCs exhibiting recurrent mutations in seven of them (*BAP1*, *ASXL1*, *SETD2*, *SMARCA4*, *DNMT3A*, *TET2*, and *WT1*), an observation which was not present for thymomas [15,23]. TETs have also been documented to exhibit the inactivation of tumor suppressor genes, such as *FHIT*, *MLH1*, and *E-cad*, through promoter hypermethylation [40]. Similarly, *MGMT* methylation is more prevalent in TCs (74%) than in thymomas (29%), especially in advanced stages [40], and *CDKN2* promoter methylation occurs in up to 25% of TCs and in 14% of thymomas [43]. In a similar study, Hirose et al. investigated the methylation status of *DAPK*, *p-16*, and *HPP1* genes alongside *MGMT* in 26 thymomas and 6 TCs, revealing aberrant DNA methylation patterns [44]. This study observed a correlation between the frequency of aberrant DNA methylation and the histological TET types, with a higher occurrence in TCs compared to thymomas [44].

An investigation of Chen et al. on global methylation levels and the promoter methylation status of tumor suppressor genes (TSG) *hMLH1*, *MGMT*, *p-16INK4a*, *RASSF1A*, *FHIT*, *APC1A*, *RARB*, *DAPK*, and *E-cadherin* in 65 TET samples indicated hypermethylation and reduced TSG expression in types B1 or higher thymomas [45]. Compared to early-stage TET, there was a decrease in global DNA methylation levels, while the expression of *DNMT1*, *DNMT3a*, and *DNMT3b* increased in advanced-stage thymomas [45].

Muguruma et al. also conducted a comprehensive examination of the methylation status of the two cancer-related genes, *MT1A* and *NPTX2*, in a total of 48 thymic tumor samples (31 thymomas, 17 TCs) and 22 paired normal tissue samples [46]. The methylation levels of the *MT1A* gene were significantly elevated in TC compared to thymoma (26.4% versus 9.5%), demonstrating exceptional sensitivity and specificity in distinguishing between TCs and thymomas [46]. Similarly, the *NPTX2* gene exhibited markedly higher methylation levels in TC compared to thymoma (38.0% vs. 17.5%), with notable sensitivity and specificity in terms of discriminating between TCs and thymomas [46]. Despite a significant increase in DNA methylation in TC compared to normal thymus, no significant distinction was observed between the DNA methylation levels of thymoma and normal thymus. Furthermore, no variations in the DNA methylation of *MT1A* and *NPTX2* were identified among the several histological types of thymoma based on the WHO histologic classification [46].

Moreover, Bi et al. made an examination of DNA methylation profiles across twelve tissues to distinguish Type A and Type B thymomas which were intricately connected with gene-expression data analysis [47]. A comprehensive total of 10,014 CpGs demonstrated distinct methylation patterns between these two thymoma types [47]. The integrative analysis disclosed 36 genes harboring differentially methylated CpG sites within their promoter regions. Notably, among these genes, 29 displayed hypomethylation concomitant with elevated expression levels, and zinc finger protein 396 and Fraser extracellular matrix complex subunit 1 emerged as particularly noteworthy, exhibiting the most substantial area under the curve [47].

The methylation analysis could also be employed for diagnostic purposes for the distinction of the various WHO subtypes [48]. To improve precision, particularly in morphological borderline cases, Gaiser et al. explored a methylation pattern-based classification [48]. In their study, an array-based DNA methylation analysis of 113 thymomas with detailed histological annotation was conducted. Unsupervised clustering and t-Distributed Stochastic Neighbor Embedding (t-SNE) analysis aligned thymoma samples with the current WHO classification. However, methylation analyses revealed a nuanced distinction within histological subgroups AB and B2, yielding two methylation classes: mono-/bi-phasic AB-thymomas and conventional/“B1-like” B2-thymomas. This emphasizes the potential of methylation-based classifications to refine diagnostic criteria, improve reproducibility, and impact treatment decisions [48].

In addition to the WHO criteria, the Masaoka-Koga Staging System plays a pivotal role in the prognosis of TETs. In a comprehensive study by Li et al., an intricate analysis encompassed TCGA 450 K methylation array data, transcriptome sequencing data, WHO histologic classification, and the Masaoka-Koga staging system [42]. The primary objective was to discern differentially expressed methylation sites distinguishing thymoma from thymic carcinoma, as well as identifying DNA methylation sites linked to the overall survival of TET patients. Employing pyrosequencing, 100 patients with TETs had four specific methylation sites (cg05784862, cg07154254, cg02543462, and cg06288355) sequenced from their tumor tissues. Examination of the TCGA dataset unveiled 5155 hypermethylated and 6967 hypomethylated CpG sites in Type A–B3 and Type C groups, respectively, with 3600 located within gene promoter regions [42]. Promoter hypermethylation led to the silencing of 134 genes, while 174 mRNAs experienced upregulation. Four specific genes (cg05784862/*KSR1*, cg07154254/*ELF3*, cg02543462/*ILRN*, and cg06288355/*RAG1*) emerged as independent prognostic factors for overall survival in TET patients [42]. The prognostic model, comprising these genes, demonstrated superior accuracy in predicting 5-year overall survival compared to the Masaoka-Koga clinical staging [42].

In addition, in a recent investigation conducted by Guan et al., 40 genes and 179 genes were identified as epigenetically upregulated and silenced, respectively. This corresponds to a total of 509 functionally methylated CpG sites differentiating thymomas from TCs, as revealed by the analysis of the TCGA dataset [49]. The methylation β-values of cg20068620 in *MAPK4* and cg18770944 in *USP51* exhibited a significant correlation with Recurrence-Free Survival (RFS). This indicates that these methylation sites have the potential to enhance the prognostic accuracy of TET recurrence beyond what can be achieved using the WHO classification alone [49].

### 2.6. Comparative Genomic Hybridization

Copy number alterations (CNAs) in different chromosomes have been extensively documented in TETs, and their occurrence is more prevalent in histological subtypes associated with aggressive behavior [43]. The reported chromosomal changes seem to allow the categorization of TETs into distinct groups, unveiling a connection between genetic discoveries and histology based on WHO classification [43]. Notably, there are shared alterations between Type AB and B2 thymoma, B2 and B3 thymoma, as well as B3 thymomas and TCs [43]. In summary, the loss in the 6q25.2–25.3 region, housing the *FOXC1* tumor suppressor gene located at 6p25.3, is evident in all TET subtypes except for Type B1 thymomas; although, the limited number of B1 thymomas in the analyzed cases may have influenced this outcome [43,50,51].

Another Comparative Genomic Hybridization (CGH) study conducted by Lee et al. involving 39 TET patients, revealed different chromosomal losses among the various WHO histological subtypes [52]. In particular, Type A thymomas displayed losses in chromosomes 2, 4, 6q, and 13; Type B1 in chromosomes 1p, 2q, 3q, 4, 5, 6q, 8, 13, and 18; Type B2 in chromosomes 1p, 2q, 3q, 4, 5, 6q, 8, 13, and 18; and Type B3 in chromosomes 2q, 4, 5, 6, 8, 12q, 13, and 18 [52]. Notably, Type A exhibited the least chromosomal abnormality, whereas Type B thymoma subtypes demonstrated overlapping loss patterns across various chromosomes [52]. Chromosome 9q gain was exclusive to Type B1, and chromosome 1q gain was present solely in Type B3 [52]. Additionally, Type AB exhibited losses in chromosomes 2, 4, 5, 6q, 8, 13, and 18 [52]. TCs exhibit gains in chromosomes 1q, 4, 5, 7, 8, 9q, 12, 15, 17q, 18, and 20, along with losses in 3p, 6, 6p23, 9p, 13q, 14, 16q, 17p [43].

Another interesting study, utilizing CGH, scrutinized 59 TETs and discerned recurring patterns of CNAs across diverse histotypes [53]. The “Genomic Identification of Significant Targets in Cancer” (GISTIC) algorithm unveiled 126 noteworthy peaks of CN aberration, encompassing 13 cancer-related genes. Specifically, the CN gain of *BCL2* and CN loss of *CDKN2A/B* emerged as the sole genes within their respective regions of CN aberration, and both were associated with an unfavorable outcome [53]. TET cell lines exhibited sensitivity to small interfering RNA (siRNA) knockdown of the anti-apoptotic molecules BCL2 and MCL1. The molecular markers *BCL2* and *CDKN2A* may hold potential value in the diagnosis and prognosis of TETs. This study provides initial preclinical evidence suggesting that dysregulated anti-apoptotic BCL2 family proteins may represent promising targets for TET treatment [53].

### 2.7. Fluorescence In Situ Hybridization

A tumor profiling study conducted by Enkner et al. using FISH reported a 38% deletion of the *CDKN2A* gene, 32% deletion of the *TP53* gene, and 8% deletion of the *ATM* gene [9]. Conversely, only one Type B3 thymoma displayed *CDKN2A* gene loss, and none of the Type A thymomas showed any deletion [9].

Chromosomal abnormalities have been documented across all histological subtypes of TETs, encompassing translocation t(15;19) and deletions in the 6p22-p25 region. A notable instance of such translocations is t(15;19)(q13:p13.1), resulting in the formation of the fusion gene *BRD4*–*NUT*, identified in undifferentiated TCs [54]. Among thymomas, one of the prevailing genetic alterations is situated in the chromosome 6p21.3, specifically at the major histocompatibility complex (MHC) locus, and extends to 6q25.2 to 25.3 [54].

Table 1 provides an overview of the genetic alterations outlined in this review, categorized according to the different WHO subtypes and the enumerated methods or subgroups of genetic alterations. It is obvious that there are significant genetic distinctions between thymomas and TCs, the latter exhibiting a higher degree of genetic dysregulation compared to thymomas.

**Table 1 ijms-25-01554-t001:** Overview of the genetic alterations in thymic epithelial tumors (TETs), categorized according to the different World Health Organization (WHO) subtypes and the enumerated methods or subgroups of genetic alterations listed in this review.

WHO Subtype of TETs	Genetic Alterations		
	NGS-Identified Gene Mutations	RNA-Seq-Detected Alterations	Chromosomal Abberations
MT	No GTF2I mutation [24] POLE [22]		Chromosome 11 inversion [22,24] Loss in the 6q25.2–25.3 region [43] YAP1-MAML2 fusion [22,24]
MN-T	GTF2I (p.Leu424His) [13]		Loss in the 6q25.2–25.3 region [43]
Type A Thymoma	GTF2I (p.Leu404His; 76–83% of cases) [12] HRAS [6,11], NRAS [6,11], TP53 [12], PCLO [12], HDAC4 [12], BCOR [12], CSF1R [12], FGF3 [12], NRAS [12], PAX7 [12], PTPRB [12], NOD1 [12]	Overexpression of C19MC miRNA cluster [29] Upregulation of tumor suppression, particularly p53 [38] Downregulation of MYC/Max [38], MYB [38], and FOXM1 [38] Overexpression of mRNAs: AXIN2 [33]	Loss in the 6q25.2–25.3 region [43] Chromosomal losses: 2, 4, 6q, and 13 [52] No YAP1-MAML2 fusion [24]
Type AB Thymoma	GTF2I (p.Leu404His; 76–83% of cases) [12] HRAS [6,11], NRAS [6,11], PCLO [12], HDAC4 [12], PBRM1 [12], PTPRB [12], ZMYM3 [12], NOD1 [12]	Overexpression of C19MC miRNA cluster [29] Downregulation of tumor suppression (p53 and TAp73a) [38] Upregulation of oncogenes (MYC/Max, MYB [38], FOXM1 [38], and E2F1 [38]	Loss in the 6q25.2–25.3 region [43] Chromosomal losses: 2 [52], 4 [52], 5 [52], 6q [52], 8 [52], 13 [52], and 18 [52]
Type B1 Thymoma	GTF2I (p.Leu404His) [12]	Downregulation of tumor suppression (p53 and TAp73a) [38] Upregulation of oncogenes (MYC/Max, MYB [38], FOXM1 [38], and E2F1 [38]	Chromosomal losses: 1p [52], 2q [52], 3q [52], 4 [52], 5 [52], 6q [52], 8 [52], 13 [52], and 18 [52] Chromosome 9q gain [52]
Type B2 Thymoma	GTF2I(p.Leu404His) [12], TP5325, BCOR [12], BRD4 [12], CSF1R [12], FGF3 [12], NRAS [12], PTPRB [12], ZMYM3 [12]	Downregulation of tumor suppression (p53 and TAp73a) [38] Upregulation of oncogenes (MYC/Max, MYB [38], FOXM1 [38], and E2F1 [38]	Loss in the 6q25.2–25.3 region [43] Chromosomal losses: 1p [52], 2q [52], 3q [52], 4 [52], 5 [52], 6q [52], 8 [52], 13 [52], and 18 [52] KMT2A-MAML2 translocations involving different combinations of exons 8, 9, 10, or 11 of KMT2A and exon 2 of MAML2 [27]
Type B3 Thymoma	GTF2I (p.Leu404His) [12] ASXL1 [10], DNMT3A [10], TP53 [12], HDAC4 [12], SMARCB1 [11], STK11 [11]	Downregulation of tumor suppression (p53 and TAp73a) [38] Upregulation of oncogenes (MYC/Max, MYB [38], FOXM1 [38], and E2F1 [38]	Loss in the 6q25.2–25.3 region [43] Chromosomal losses: 2q [52], 4 [52], 5 [52], 6 [52], 8 [52], 12q [52], 13 [52], and 18 [52] Chromosome 1q gain [52] Deletion in CDKN2A gene [9] KMT2A-MAML2 translocations involving different combinations of exons 8, 9, 10, or 11 of KMT2A and exon 2 of MAML2 [27]
TC	GTF2I (p.Leu404His; only 8% of cases) [12] TP53 [9,12,15], BCOR [10], HRAS [12,15], CYLD [12,15], PCLO [12], HDAC4 [12], PBRM1 [12], BRD4 [12], CSF1R [12], FGF3 [12], NRAS [12], PAX7 [12], NOD1 [12], CDKN2A [17], BAP1 [17], PBRM1 [17], KIT [18], DDR2 [18], PDGFRA [18], ROS1 [18], IGF1R [18], ALK [9,16], ATM [9], ERBB4 [9], PI3K [19], MYO16 [20], BAP1 [21], ABI1 [21], BCL9L [21], CHEK2 [21], ZNF429 [21], ZNF208 [21], GNAQ [21], TET2 [15], SETD2 [15], FBXW7 [15], RB1 [15] Pathogenic somatic SNVs in TP53 [16], ERBB2 [16], KIT [16], and KRAS [16] Genome copy number gains at KIT [15], and AHNAK2 [15] Hypermethylated genes: BAP1 [15,23], ASXL1 [15,23], SETD2 [15,23], SMARCA4 [15,23], DNMT3A [15,23], TET2 [15,23], WT1 [15,23] MGMT promoter methylation (74%) [40] CDKN2 promoter methylation (25%) [43]	C14MC miRNA cluster [9] Upregulation of miR-21 [9], miR-9-3 [9], and miR-375 [9] Downregulation of miR-34b [9], miR-34c [9], miR-130a [9], and miR-195 [9] Downregulation of tumor suppression (p53 and TAp73a) [38] Upregulation of oncogenes (MYC/Max, MYB [38], FOXM1 [38], and E2F1 [38]	Loss in the 6q25.2–25.3 region [55] Chromosomal losses: 3p [43], 6 [43], 6p23 [43], 9p [43], 13q [43], 14 [43], 16q [43], 17p [43] Chromosomal gains in 1q [43], 4 [43], 5 [43], 7 [43], 8 [43], 9q [43], 12 [43], 15 [43], 17q [43], 18 [43], and 20 [43] Deletion in CDKN2A gene [9] Fusion gene BRD4-NUT [54] TP53 deletions [9] ATM deletions [9]

## 3. Signaling Pathways and Integrative Analysis in Thymic Epithelial Tumors

In addition to investigations into genetic alterations, pathway analyses were conducted using various laboratory assays and bioinformatics tools to improve our understanding of the pathogenesis of TETs.

### 3.1. General Findings in Pathway Analyses in Thymic Epithelial Tumors

Meng et al. conducted a study characterizing genes and their products in TETs through “Gene ontology” (GO) analysis, focusing on cellular components, molecular functions, and biological processes, as well as pathway analyses [56]. Notably, dysregulation was observed in processes such as MHC class II protein complex assembly and was associated with peptide antigen, calcium-activated phosphatidylcholine scrambling, and the release of cytoplasmic sequestered NF-κB [56]. Concurrently, downregulation was identified in pathways like the intestinal immune network for immunoglobulin A production, cytokine–cytokine receptor interaction, the calcium signaling pathway, and those associated with autoimmune diseases [56].

Evaluation of 121 samples within the TCGA datasets unveiled 297 DEGs, all of which belonged to the category of downregulated genes, whereas a further examination of signaling pathway enrichment using the “Kyoto Encyclopedia of Genes and Genomes” (KEGG) database pinpointed 154 DEGs that exhibited significant associations with 16 distinct signaling pathways [57]. The top five KEGG signaling pathways included hsa03320 (PPAR signaling pathway), hsa04923 (regulation of lipolysis in adipocytes), hsa04152 (AMPK signaling pathway), hsa00360 (phenylalanine metabolism), and hsa00980 (metabolism of xenobiotics by cytochrome P450) [57]. The shared enrichment pathways for driver genes at the intersection of the TC and thymoma clusters by Yang et al. comprised lysine degradation (hsa00310), endocrine and other factor-regulated calcium reabsorption (hsa04961), long-term depression (hsa04730), long-term potentiation (hsa04720), gastric acid secretion (hsa04971), and the ErbB signaling pathway (hsa04012) [21].

Different TET subtypes exhibit distinct gene alterations, consequently implicating diverse signaling pathways. Lai et al. identified 204 proteins with differential expression across three TET subtypes (AB, B2, and B3) [58]. The utilization of “Ingenuity Pathway Analysis” (IPA) was applied to identify pathways with increased representation in every category [58]. IPA, recognized as the most extensive knowledge repository and analytical platform, provides insights into the most profoundly influenced signaling and metabolic pathways, molecular networks, and biological processes within a given dataset [59]. In alignment with the “hallmarks of cancer” postulated by Hanahan and Weinberg in 2000 [60], which encompass six key characteristics of malignancies, Lai et al.’s study’s pathway analysis unveiled canonical pathways associated with signal transduction, nuclear receptor activation, and the complement system across the three TET types [58].

In a study by Liang et al., it was illustrated that TCs exhibited an enriched functionality in various pathways, encompassing cell adhesion, cell migration, cell differentiation, immune system processes, immune response, and nervous system development [31]. The highly expressed genes in cluster TCs showed enrichment in pathways such as the PI3K/Akt signaling pathway [31].

Additionally, Radovich et al. computed 10 pathway scores and created clusters that exhibited a statistically significant association with the histological subtypes [38]. The analysis revealed that cluster 2 exhibited notably low activity in cell cycle, apoptosis, TSC/mTOR, and core reactive pathways. In contrast, cluster 1 demonstrated elevated Epithelial-Mesenchymal Transition (EMT) activity [38]. Meanwhile, clusters 3 and 4 displayed markedly increased activity in cell cycle, hormone signaling, and TSC/mTOR pathways, coupled with diminished Ras/MAPK and breast reactive activity [38].

*GTF2I*, a thymoma-specific oncogene, encodes the transcription factor TFII-I/BAP-135, pivotal for interpreting diverse signals and regulating transcription [61]. Activated by stimuli from T- and B-cell receptors or growth factor pathways, TFII-I influences the transcription of specific genes, such as FOS and cyclin D1 [61]. The distinctive GTF2I Leu424His mutation identified in TETs disrupts TFII-I degradation, resulting in the upregulation of pathways associated with cell proliferation, receptor signaling, and morphogenesis, particularly in the WNT and HH signaling pathways. In contrast, pathways related to apoptosis, the cell cycle, DNA-damage response, hormone receptor signaling, breast hormone signaling, Ras/MAPK, RTK, and mTOR are downregulated [61].

### 3.2. C-KIT, ErbB/EGFR, IGF-1R, Ras/MAPK, PIK3/Akt/mTOR, and p53 Signaling Pathways

The ErbB/EGFR, IGF-1R, Ras/MAPK, PI3K/Akt/mTOR, and p53 signaling pathways play a crucial role in TET pathogenesis. Figure 4 illustrates a simplified representation of these pathways and their interactions.

In terms of gene pathways, p53 in TETs, which encompasses *TP53* and *ATM*, exhibited the highest frequency of alterations (20.4%) in a study by Sakane et al., trailed by the receptor tyrosine kinase (RTK)/Ras pathway (18.5%) and the PI3K pathway (5.6%) [62].

NGS examination of TETs has revealed various SNVs within genes associated with the p53, AKT, MAPK, and KRAS signaling pathways [16]. A notable increase in the presence of mutated genes related to the Ras and PI3K/Akt signaling pathways has been documented (*AKT3*, *ALK*, *CSF1R*, *FGFR4*, *KRAS*, *NRAS*, *HRAS*, *PIK3CA*), indicating a potential involvement of this pathways in the development of TETs [16]. In the same context, Radovich et al. were able to demonstrate that the overexpression of miRNA cluster C19MC in Type A und AB thymomas induces the activity of the PI3K/Akt/mTOR pathway [29].

Similarly, Psilopatis et al. conducted a network analysis of the regulatory axes involving lncRNA, mRNA, and miRNA, revealing an upregulated cluster of miRNAs in thymomas [63]. This cluster initiated the expression of target protein-coding genes, leading to the perturbation of diverse biological pathways, including the PI3K/Akt/mTOR, FoxO, and HIF-1 signaling pathways [63]. Moreover, the lncRNAs ADAMTS9–AS1, HSD52, LINC00968, and LINC01697 were identified as potential indicators for accurate patient categorization, distinguishing between low- and high-risk TETs, and providing effective predictions of recurrence probability [63]. Similar observations were made by Ji et al.’s network analysis of the regulatory axes involving lncRNA-mRNA-miRNA, who identified a set of 1695 miRNAs that are upregulated in thymomas [26]. These have the potential to induce the expression of target protein-coding genes, resulting in the disruption of multiple biological pathways, including the PI3K/Akt signaling pathway, FoxO signaling pathway, and HIF-1 signaling pathway [26]. Within this network, the overexpressed miRNA hsa-let-7a-3 demonstrates interactions with eight protein-coding genes (*INSR*, *IGF1*, *IL10*, *IGF1R*, *ITGB3*, *COL5A2*, *ZNF322*, *PXDN*, *TGFBR1*) and has the potential to augment their expression [26].

Although activation of the PI3K/Akt pathway is observed in many cancers, the extent of activation differs significantly [29]. In numerous cases, this activation is accomplished via somatic mutations in *PIK3CA* or *PTEN* [29]. In these terms, the IPA has disclosed a notable upregulation of PI3K in Type A and AB thymomas, alongside *PREX2* (a *PTEN* antagonist), *RAS*, and *MAGI* [29]. Moreover, there is significant downregulation of the FOXO family of transcription factors, recognized to be inactivated when the PI3K/Akt pathway is activated, along with BIM, p21cip1, and MAST2 [29].

Maury et al. investigated the protein expression and activation status of crucial elements in the Akt/mTOR pathway, specifically Akt, mTOR, and P70S6K, in 11 thymomas of Types A, B, and AB, as well as in normal thymus [64]. Unlike normal thymus, where only Akt and phospho-Akt were expressed, thymomas displayed activation in the Akt and mTOR protein, which was observed in the phosphorylated form [64]. Phospho-P70S6K was also present in all thymic tumors, regardless of subtype, and was absent in normal thymus [64]. The data indicates the increased activity of the Akt/mTOR pathway [64]. Alterations in genes that code for regulatory subunits of PIK3 have been detected in a tumorigenic TC cell line [64]. Interestingly, genomic alterations in TP53/CDK, EGFR/Ras, and PI3K/mTOR pathways have also been documented in metastatic TETs [65]. All these observations underscore the significance of the PIK3/Akt/mTOR pathway in TETs.

Moreover, it is imperative to recognize that alterations within the ErbB/EGFR signaling pathway have emerged as crucial determinants of tumor behavior, offering promising prospects as diagnostic and prognostic markers. According to a recent investigation, ErbB/EGFR overexpression is associated with tumor aggressiveness [21]. The study observed an abundance of ErbB and T-cell signaling pathways in TC, while the thymoma group showed significant pathways related to longevity regulation and central carbon metabolism in cancer [21]. This was also confirmed in another examination of prognostic factors, where cases of 44 TCs revealed a noteworthy finding [66]. The overall survival of patients with EGFR pathway mutations was significantly shorter compared to those without, as indicated by a univariate analysis. Further reinforcing this observation, EGFR pathway mutations emerged as an independent factor associated with poor overall survival in a subsequent multivariate analysis [66].

On the contrary, a different study conducted by Girard et al. scrutinized 45 thymic tumors, comprising 38 thymomas (8 Type A, 22 Type B2, 8 Type B3) and 7 TCs) [67]. Their profiling of these tumors aimed to identify mutations in genes associated with the EGFR signaling pathway, particularly those known for recurrent nucleotide mutations in human cancers. Notably, no mutations were identified in the *EGFR* kinase domain or in any other examined genes within the EGFR signaling pathway [67].

The IGF pathway governs various biological processes, including metabolism and cell growth. IGF I and II bind to Insulin-like Growth Factor 1 Receptor (IGF-1R), a heterotetrameric transmembrane glycoprotein with an intracellular tyrosine kinase domain [61]. The interaction between IGF and IGF-1R is regulated by IGF binding proteins (IGFBPs 1–6), activating two major pathways: the insulin receptor substrate (IRS)/PI3K/Akt/mTOR pathway, which is primarily associated with metabolic effects, and the SHC/Ras/MAPK pathway, which is primarily linked to mitogenic effects. IGF-1R expression is evident in all histological subtypes of TETs, particularly in aggressive subtypes and those at an advanced disease stage [61]. Moreover, TETs frequently exhibit a loss of heterozygosity of Insulin-like Growth Factor 2 Receptor (IGF-2R), potentially leading to compensatory upregulation of IGF-1R [61,68].

IGF-1R has been commonly observed to be overexpressed in TCs [69]. Despite not being recognized as a prominent driver of molecular alteration, the activation of IGF-1R plays a role in various processes associated with oncogenesis. The resistance to EGFR inhibitors has been linked to the activation of IGF-1R [69]. This interaction results in continuous stimulation of the PI3K/Akt pathway and the suppression of the pro-apoptotic protein surviving [69].

The proto-oncogene *KIT*, located at chromosome 4q12, encodes a type III receptor tyrosine kinase known as c-KIT (CD117), which plays a pivotal role in diverse cellular processes [61]. Upon interaction with its ligand and stem cell factor (SCF), c-KIT undergoes dimerization, which is the autophosphorylation of tyrosine residues, and activation of protein kinases. This initiates multiple signal transduction pathways, including MAPK, PI3K/Akt/mTOR, PLCγ/DAG/IP3, JAK/STAT, and SRC, resulting in the activation of cell survival, proliferation, motility/invasion, and angiogenesis [61]. TCs frequently exhibit c-KIT overexpression (46–79%), with *KIT* mutations detected in less than 10% of cases. Conversely, c-KIT overexpression is rare in thymomas (2–4%), and apart from a documented *KIT* deletion in a patient with AB Thymoma in the TCGA PanCancer Atlas, no other mutations have been reported [61].

These data were also confirmed by Prays et al. who observed a c-KIT overexpression in 50–88% of TCs, and that overexpression has been linked to a poorer prognosis [54]. Elevated *KIT* expression has been suggested as a characteristic of WHO Type A and AB thymomas, where it potentially plays a pivotal role by triggering the MAPK signaling pathway [70]. Yang et al. demonstrated that the activation of *KIT* leads to a cascade of abnormal changes in mRNA, miRNA, and DNA methylation. The authors utilized miRNA data from TCGA to explore its association with KITLG expression, identifying 79 positively and 78 negatively correlated miRNAs [70]. Some of these miRNAs have prior associations with thymoma or the thymus [70]. Upregulated miRNAs, like miR-125a and miR-34a, may impact inflammatory pathways and thymoma cell differentiation [70]. Conversely, downregulated miRNAs such as miR-106, targeting *MEK2*, could affect thymic immune function, and are linked to the upregulated MAPK signaling pathway in Type A and AB thymoma. MiR-363 and miR-20b act as tumor suppressors in thymoma development [70].

### 3.3. Notch, Hedgehog, and Wnt Signaling Pathways

The Notch, Hedgehog (HH), and Wnt/β-catenin pathways play a crucial role in the development of both vertebrates and invertebrates and can be reactivated in various cancers [71]. These pathways are also implicated in thymus development, with paracrine signaling between thymic stroma and hematopoietic cells [72,73]. Figure 5 depicts a simplified representation of the Notch, Hedgehog (HH), and Wnt/β-catenin pathways [74].

Notch plays a crucial role in T-cell development within the thymus. Although its function in normal thymic epithelium is not completely understood, the activation of the pathway has been documented [72,73]. Moreover, Notch participates in short-range intercellular communication through interaction with ligands on adjacent cells [74]. The binding of ligands initiates extracellular cleavages (S2 and S3) facilitated by ADAM10, ADAM17, and γ-secretase, resulting in the liberation of the Intracellular domain (ICN). ICN then moves to the nucleus, where it partners with RBPJ/CSL, altering the repressor complex to a coactivator complex. This transformation activates the transcription of target genes [74].

Riess et al., for example, hypothesized that the expressions of *GLI1*, *CTNNB1*, and *NOTCH1* would be increased in the thymic tissue microarray (TMA) owing to the importance of the Shh, Wnt, and Notch pathways in thymic development [71]. However, according to their findings, *GLI1*, *CTNNB1*, and *NOTCH1* did not show a significant increase in thymomas when compared to benign thymic tissue [71].

The frequent translocations involving fusion genes such as *KMT2A*–*MAML2*, as well as others like *HADHB*–*REEP1*, *COQ3*–*CGA*, and *MCM4*–*SNTB1* identified by Ji et al. are all implicated in diverse pathways [26]. The fusion gene *KMT2A*–*MAML2* has been observed to suppress the promoter activation of the *NOTCH1* target gene *HES1*, exhibiting oncogenic characteristics [26].

The HH signaling pathway is a vital and evolutionarily conserved mechanism that regulates organ formation during embryonic and postnatal development. It plays a key role in controlling cellular processes in adults, including proliferation, tissue differentiation, and tissue repair. Components of the pathway include *SHH*, *IHH*, *DHH* ligands, as well as PTCH, SMO, SUFU, and GLI transcription factors [74]. In the absence of ligands, Ptch1 inhibits SMO accumulation in primary cilia, blocking pathway activity. Ligand binding induces *PTCH1* internalization and degradation, allowing *SMO* to enter cilia, leading to *SUFU*–*GLI* complex dissociation and *GLI* activation in the nucleus. This activation triggers the expression of Hedgehog target genes like *GLI1* and *PTCH*. Besides the canonical pathway, non-canonical mechanisms have been implicated in cancer development [74]. Continuous Shh signaling is particularly linked to the development of multiple malignancies [71].

The initiation of the canonical Wnt signaling pathway occurs through the binding of *Wnt* to the Frizzled (Fz) receptor and the LRP5/LRP6 co-receptor [74]. This interaction triggers the activation of the pathway, involving key components such as *Dsh* and β-catenin *GSK3β*. Axin binds to the intracellular domain of LRP5/6, leading to the phosphorylation of β-catenin by *GSK3* and *CK-1γ* [74]. In its inactive state, *GSK3* phosphorylates β-catenin, targeting it for ubiquitination and subsequent degradation. This mechanism prevents β-catenin from entering the nucleus, where it would normally activate the expression of target genes. However, upon *Wnt* binding, the disruption of the “destruction complex” occurs, inhibiting the phosphorylation of β-catenin [74]. The accumulation of hypo-phosphorylated β-catenin in the cytosol enables its translocation to the nucleus. In this nuclear environment, it interacts with TCF/LEF transcription factors, thereby regulating gene expression and influencing processes such as cell differentiation and proliferation [74].

In an investigation, Chen et al. explored the potential involvement of *WNT4* and *FOXN1* in the development of TETs [75]. In thymus development, β-catenin stabilizing mutations hinder both development and function [76], and thymic epithelial cells transmit *WNT* signals to maturing thymocytes [77]. According to this study, decreasing *WNT4* and *FOXN1* could affect the apoptotic processes of TET cells and the growth of tumors in nude mice [75], suggesting that the Wnt signaling pathway is involved in regulating the maturation of thymic epithelial cells by modifying the activity of *FOXN1* [75]. The *FOXN1* gene exhibited a significant downstream impact through its regulatory influence on *WNT4* with an increase in aggressive behavior [75].

The role of the Wnt pathway is also confirmed by Vodička et al. who examined the mRNA expression levels of genes associated with the Wnt pathway and E-cadherin, showing a significant increase in both Type A and B thymomas [33]. The expression levels increased gradually from Type B1 to B3 and were higher in advanced disease stages. In recurring Type B thymomas, there was a notable elevation in the mRNA levels of these genes [33]. Even with the activation of the Wnt pathway in less aggressive Type A thymomas, the negative feedback mechanism of the pathway was maintained by overexpressing the inhibitory molecule *AXIN2*, a phenomenon not observed in Type B thymomas [33]. These results suggest that the Wnt pathway is activated in human thymomas and may contribute to oncogenesis [33]. Interestingly, mutations in *GTF2I* have been linked to the overexpression of the Wnt and HH signaling pathways [38]. Certain earlier studies have suggested a connection between thymomas of Type B and the loss of heterozygosity of the *APC* gene [33], which plays a crucial role in regulating the canonical Wnt signaling pathway, hindering the buildup of β-catenin in the cytoplasm [33].

Yang et al. delved deeper into the enriched signaling pathways within subgroups of TETs, conducting a comparative analysis through GO and KEGG clustering, which revealed distinct enrichment of Wnt, MAPK, HH, AMPK, and cell junction assembly signaling pathways in TETs other than B3 [20] (Figure 6A–C). B3 thymomas were specifically linked to the lysine degradation pathway (Figure 6B) [20].

Figure 7 visualizes the markedly enriched biological processes and pathways in TCs and thymomas [20]. Both groups share covalent chromatin modification and histone modification pathways [20]. In contrast, thymomas exhibit distinct enrichment in signaling pathways, including MAPK, Wnt, AMPK, Notch, HH, and cell junction assembly (Figure 7A–C). Conversely, TCs are characterized by exclusive enhancement related to extracellular matrix-receptor (ECM-receptor) interaction, positive regulation of the cell cycle process, and activation of innate immune response pathways (Figure 7B–D) [20].

Moreover, Figure 8 illustrates that the homophilic cell adhesion through the pathway involving plasma membrane adhesion molecules was common to both B3 thymoma and TCs. However, the distinctive pathways enriched in B3 thymomas (Figure 8A–C) and TCs (Figure 8B–D) exhibited variations [20]. 

### 3.4. Comparison between Thymoma-Associated Myasthenia Gravis and Non-Myasthenia Gravis Thymoma

Several studies suggest different pathway activation between TAMGs and NMGs. A KEGG analysis identified significant upregulation in the TGF-beta and HTLV-I signaling pathways in TAMG compared to NMG thymomas [30]. Subsequent qRT-PCR analysis confirmed upregulation of *CCL25* and downregulation of *MYC*, *GADD45B*, and *TNFSF12* in TAMGs. *CCL25*, crucial in T-cell development, showed marked elevation in 80% of TAMG cases. Only one NMG case exhibited *CCL25* overexpression, implying it had a potential role in MG development [30].

Yamada et al. report additional findings from a KEGG analysis (*n* = 310) of pathways related to the TGF-beta signaling pathway: 19 functional pathways (6%) exhibited an MG association in AB thymomas; 10 pathways, including those associated with Oxidative phosphorylation, Parkinson’s disease, and Alzheimer’s disease, were notably upregulated; and 9 pathways, such as those related to Adherens junction, AGE-RAGE signaling and TGF-beta, showed significant downregulation in TAMG compared to NMG Type AB thymomas [78]. In the case of Type B2 thymoma, only eight functional pathways demonstrated an MG association: the pathway associated with Olfactory transduction was significantly upregulated in TAMG cases, while seven pathways, including Protein processing, metabolism, and DNA replication, exhibited downregulation in TAMG B2 thymomas. Importantly, the specific upregulated and downregulated pathways were distinct and did not overlap between Type AB and B2 thymomas [78].

Additionally, a recent study by Benítez et al. has provided novel insights into MG activation pathways in TETs. TAMGs have inhibited pathways related to angiogenesis, EMT, extracellular matrix organization, cell adhesion, motility, and tyrosine kinase receptor signaling [79]. MG was associated with the activation of the oxidative metabolic pathway, and no changes were observed in immunity-related pathways [79]. MG gene-expression profiles resembled those of favorable TET subtypes (A, AB, B1). Interestingly, Type AB and B1 thymomas showed the suppression of interferon γ and α pathways, while Type B2 thymomas had the matrix organization pathway suppressed [79]. In contrast, Type B3 thymomas exhibited the suppression of pluripotential cell regulation, and TCs had activated EMT and NF-kappaB pathways, indicating biological aggressiveness. In summary, TAMGs have suppressed invasion and metastatic pathways, with no specific activation pathways identified for autoimmune diseases [79].

Moreover, MiR-20b, which was downregulation in TAMG was detected by Xin et al., functions as an anti-oncogene in the progression of thymoma and TAMG [35]. Its role in suppressing tumors in thymoma is probably linked to hindering NFAT signaling by suppressing the expression of *NFAT5* and *CAMTA1* [35].

Table 2 illustrates the distinctive genetic characteristics of TAMGs and their interactions with signaling pathways.

## 4. Clinical Implications and Target Therapy in Thymic Epithelial Tumors

The primary approach to treating thymoma currently involves surgical resection accompanied by postoperative radiotherapy and chemotherapy [21]. Despite these applications, TC has a substantial risk of recurrence and mortality [21]. Currently, there is no firmly established targeted therapy for TETs, attributed to the limited number of recurrent and actionable mutations, as well as the rarity of TETs, making clinical trials challenging [12]. Furthermore, the insufficient comprehension of genomic alterations in TETs has impeded the progress in developing targeted therapy for TETs [11]. Importantly, TS has not revealed genomic mutations linked to oncogenic drivers, specifically in terms of drug sensitivity [80].

### Target Therapy in Thymic Epithelial Tumors

Nevertheless, progress in understanding the molecular changes and pathways relevant to TET pathogenesis has promoted the evaluation of potential targeted-treatment options. Encouraging results have been documented concerning the effectiveness of recently developed targeted therapies for patients with TETs, especially those carrying distinct genetic modifications [16]. These therapeutic approaches comprise anti-angiogenic drugs (including VEGR, FGFR, PDGFR), poly (ADP-ribose) polymerase (PARP) inhibitors (like BAP1), cyclin-dependent kinases (CDK, RB), inhibitors targeting receptor tyrosine kinases (such as KIT, IGF1R), and PI3K/mTOR inhibitors (PI3K) [16].

Various phase I/II studies on mTOR inhibitors, e.g., rapamycin, have been conducted in advanced TETs demonstrating high rates of disease control [64]. Clinical trials have also investigated everolimus, sunitinib, and lenvatinib as potential treatments for patients facing refractory or recurrent TETs [11]. Regarding angiogenesis inhibitors, a phase II trial combining bevacizumab with erlotinib did not show any tumor response [69], whereas a phase I study with docetaxel and aflibercept, a VEGF-A-binding soluble receptor, resulted in a partial response in one thymoma patient [69].

Furthermore, the effectiveness of therapy targeting MCL-1 and BCL-xL inhibition has been documented [11]. Petrini et al. demonstrated that cell lines derived from TETs exhibited sensitivity to siRNA knockdown targeting the anti-apoptotic molecules BCL2 and MCL1 [53]. Exploration of activating mutations within the *KIT* oncogene, identified in around 5% of TCs, has been undertaken as a potential therapeutic target for tyrosine kinase inhibitors (TKIs) like imatinib and sunitinib, yielding encouraging results in specific patients [12,15]. While around 80% to 86% of TCs express the KIT protein, this expression does not necessarily correlate with their response to TKIs [12]. Hence, only TCs with an activating *KIT* mutation are considered potential candidates for TKI therapy [12].

Moreover, combining the TKI with ABT263, an inhibitor targeting BCL2/BCL-XL/BCL-W, resulted in diminished proliferation in TET cells when combined with the tyrosine kinase inhibitor sorafenib which is known for downregulating MCL1 [53]. Another issue is that patients with metastatic thymomas harboring *BRCA2* mutations might experience advantages from the inhibition of PARP [81].

Alongside the infrequent occurrence of *EGFR* activating mutations and the presence of *RAS* mutations in thymic tumors, the limited responsiveness to EGFR inhibitors is well-accounted for. In a phase II trial involving chemo refractory thymic tumors, gefitinib exhibited a partial response in only one out of nineteen patients with thymoma and stable disease in fourteen patients, including seven TCs [69]. Notably, cetuximab demonstrated partial response in recurrent thymomas after extensive pretreatment, all of which expressed robust EGFR [69]. Figitumumab, a monoclonal antibody targeting IGF1-R, demonstrated clinical efficacy in a thymoma patient resistant to conventional treatments [69].

The potential disruption in the cyclin-dependent kinase pathway could be mitigated utilizing cyclin-dependent kinase inhibitors like milciclib [43]. A phase II study assessed the effectiveness of milciclib in TC (NCT01011439) [43]. To date, 43 individuals have been registered and subjected to treatment, comprising 26 diagnosed with TC and 9 presenting with B3 thymoma [43]. Among the 30 patients available for evaluation, 14 successfully reached the specified primary outcome of Progression-Free Survival (PFS) at 3 months and one manifested a partial response [43]. Notably, clinical investigations have indicated notable antitumor effectiveness of inhibitors targeting histone deacetylase, implying they have potential as efficacious agents for epigenetic therapeutics in TETs [63]. Nevertheless, additional studies involving more extensive patient groups are necessary to confirm the clinical efficacy and safety of innovative epigenetic agents for managing patients with TETs [63].

## 5. Challenges and Future Perspectives in Thymic Epithelial Tumors

The therapeutic landscape for TETs grapples with impediments that hinder the formulation of effective treatment strategies. A prominent challenge lies in the limited recurrence of mutations within TETs, rendering the identification of therapeutic targets a formidable task [12]. The challenges are compounded by the ongoing struggle to identify oncogenic drivers and actionable mutations in thymomas through TS. As TETs are uncommon, their genomic characteristics have not been extensively explored in comparison to more prevalent cancers like lung cancer and malignant melanoma. Tumor mutational profiling has revealed occasional activating mutations in tyrosine kinases that can be targeted, but these mutations have not been effectively addressed beyond isolated case reports [38]. To enhance our understanding in the pathogenesis of TETs, further basic science studies are required to explore various hitherto underexplored signaling pathways in TETs. This includes pathways such as MAPK and ErbB, as well as the Hippo signaling pathway [20,82]. Furthermore, the rarity of TETs presents a substantial obstacle to conducting comprehensive clinical trials, constraining the generalizability of findings, and impeding the establishment of robust treatment protocols [82]. Due to the rarity of these tumors, acquiring a substantial number of cases for drawing significant and meaningful conclusions is challenging [83]. Advancements in single-cell genomics could aid in better understanding the heterogeneity within thymomas by capturing genetic alterations at the cellular level. Overall, the discussion on the challenges and progress of sample acquisition and analysis underscores the necessity for a comprehensive approach to thymoma research.

In a global context, fostering international collaborations and data-sharing initiatives becomes paramount. The pooling of resources and data from diverse regions can contribute to the accumulation of larger and more diverse patient cohorts, thereby facilitating more robust clinical trials and advancing research efforts. To navigate these challenges, a comprehensive and collaborative approach becomes imperative for unlocking breakthroughs in the realm of TET diagnosis and therapeutics.

## 6. Conclusions

In conclusion, the management of TETs, encompassing thymoma and TC, is a complex task due to their rarity and heterogeneity [1,3]. The reliance on histology alone proves inadequate in predicting the biological behavior of these tumors [1,3]. Molecular pathology, specifically NGS, emerges as a pivotal tool, which could provide a comprehensive understanding of the genetic alterations and signaling pathways crucial for precise diagnosis, prognosis, and potential therapeutic strategies [14]. This review underscores the necessity for continued research into molecular pathways to enhance our nuanced understanding of thymic tumor biology, contributing to improved clinical outcomes [1,82].

The complex genetic and epigenetic landscape of TETs is progressively being unraveled through diverse molecular studies, providing potential diagnostic and prognostic markers. Due to the challenges posed by the rarity of these tumors, collaborative international efforts are essential for unlocking breakthroughs in TET diagnosis and therapeutics. The amalgamation of findings from diverse molecular techniques and the exploration of targeted therapeutic strategies could offer us a comprehensive approach to address the challenges associated with TETs, ultimately contributing to improved clinical outcomes.

## Figures and Tables

**Figure 1 ijms-25-01554-f001:**
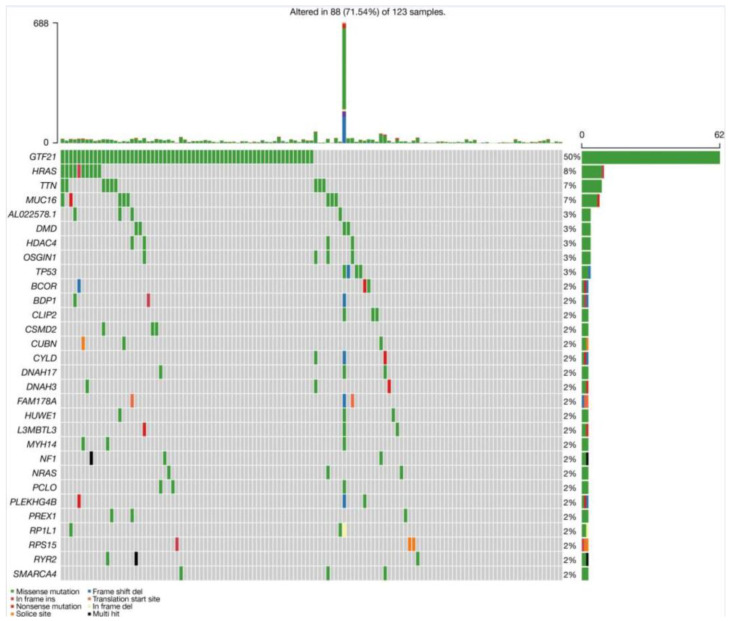
Analysis of the genetic-alteration landscape of somatic mutations in “The Cancer Genome Atlas thymoma” (TCGA-THYM) database ([21], License Number: CC BY-NC-ND 4.0).

**Figure 2 ijms-25-01554-f002:**
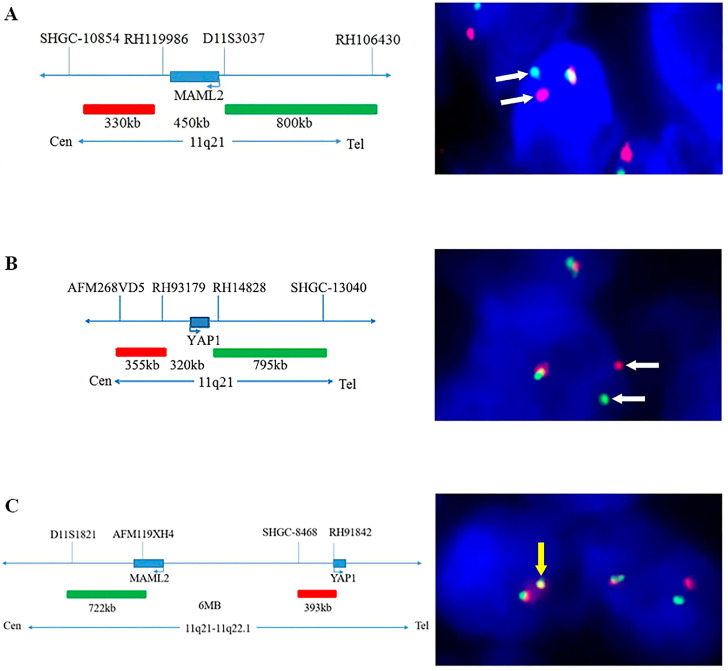
The features of the probes and illustrative outcomes from the Fluorescence In Situ Hybridization (FISH) analysis are provided as follows: (**A**) description of the dual-color break-apart probe *MAML2* and its labeling; (**B**) description of the dual-color break-apart probe *YAP1* and its labeling; (**C**) description of the fusion probe *YAP1*–*MAML2* and its labeling. The representative positive cells in the FISH analysis are also detailed. In the visual representation, separate signals are indicated with white arrows, while fusion signals are denoted with yellow arrows ([22], License Number: CC BY-NC 4.0).

**Figure 3 ijms-25-01554-f003:**
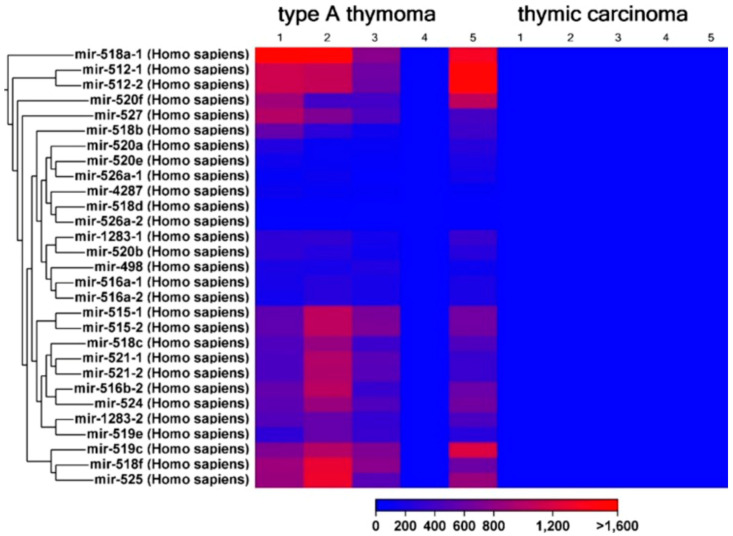
Heatmap illustrating overexpression of the C19MC microRNA (miRNA) cluster in four out of five Type A thymomas, while thymic carcinomas (TCs) exhibit minimal expression, with a false discovery rate *p*-value < 0.0005 ([29], License Number: CC BY 4.0).

**Figure 4 ijms-25-01554-f004:**
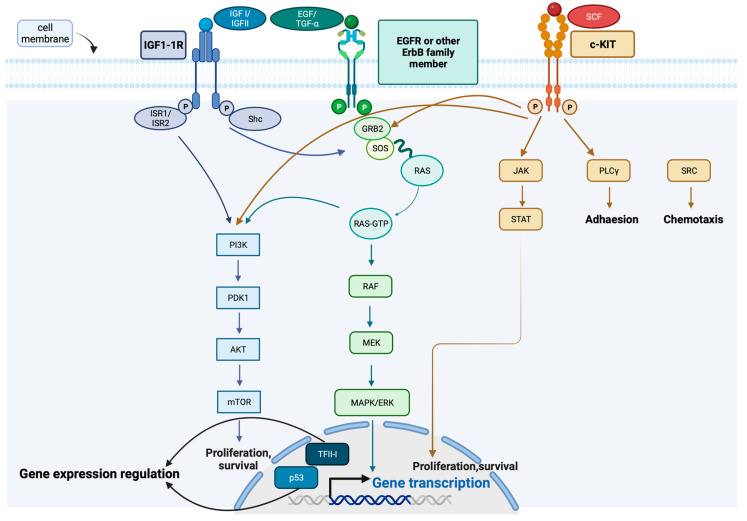
Simplified illustration of the c-KIT, ErbB/EGFR, IGF-1R, Ras/MAPK, PIK3/Akt/mTOR, and p53 signaling pathways (Created with BioRender.com).

**Figure 5 ijms-25-01554-f005:**
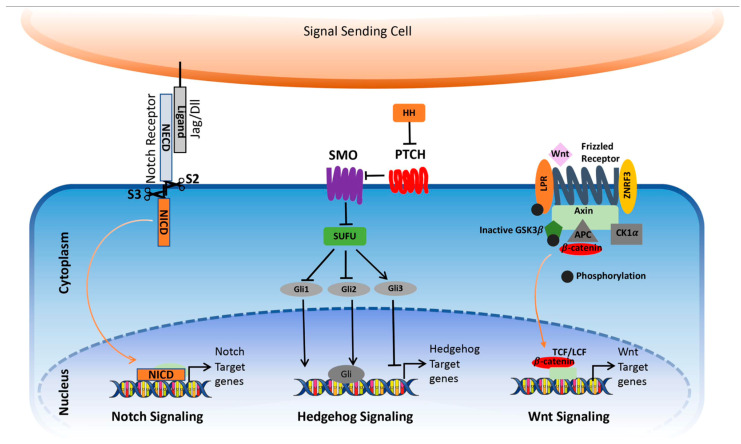
Streamlined depiction of the classic signaling pathways in cancer, including Notch, Hedgehog (HH), and Wnt//β-catenin ([74], License Number: CC BY).

**Figure 6 ijms-25-01554-f006:**
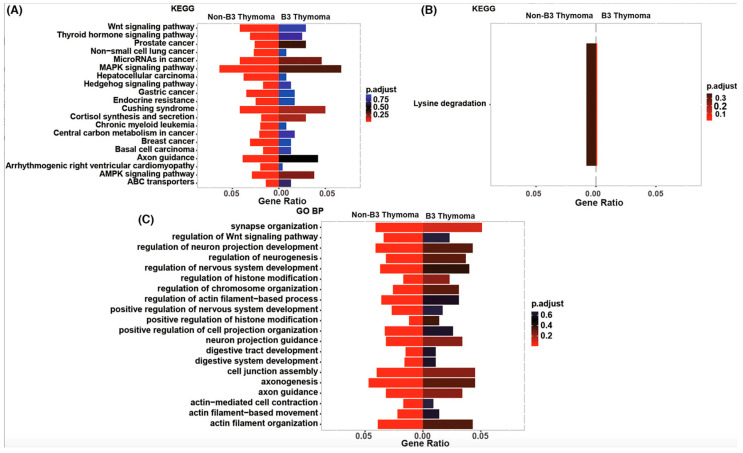
Exploration and comparison of enriched signaling pathways in thymic epithelial tumor (TET) subgroups through “Gene Ontology” (GO) and “Kyoto Encyclopedia of Genes and Genomes” (KEGG) clustering: (**A**) KEGG analysis of Non-B3 and B3 thymoma; (**B**) exclusive enrichment of the lysine degradation pathway in B3 Thymoma; (**C**) “Genome Ontology Biological Process” (GO BP) analysis of Non-B3 and B3 thymoma ([20], Licence: https://creativecommons.org/licenses/by/4.0/, accessed on 11 January 2024).

**Figure 7 ijms-25-01554-f007:**
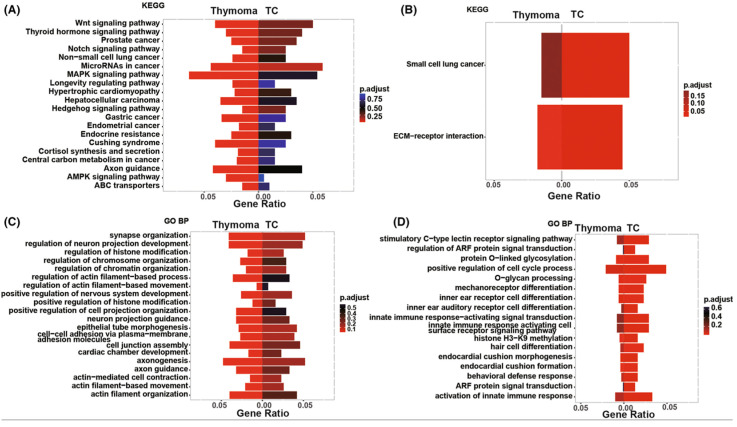
Comparison of enriched pathways in thymomas and TCs: (**A**) pathways exclusively enriched in thymomas according to KEGG analysis; (**B**) pathways exclusively enriched in TCs according to KEGG analysis; (**C**) pathways exclusively enriched in thymomas according to GO BO analysis. (**D**) Pathways exclusively enriched in TCs according to GO BO analysis ([20], Licence: https://creativecommons.org/licenses/by/4.0/, accessed on 11 January 2024).

**Figure 8 ijms-25-01554-f008:**
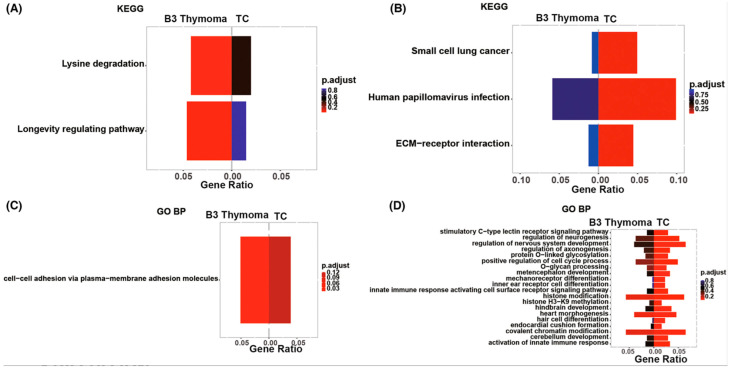
Contrast of enriched pathways in B3 Thymoma and TCs: (**A**) pathways exclusively enriched in B3 thymoma based on KEGG analysis; (**B**) pathways exclusively enriched in TCs according to KEGG analysis; (**C**) pathways exclusively enriched in B3 thymoma as per GO BP analysis; (**D**) pathways exclusively enriched in TCs through GO BP analysis ([20], Licence: https://creativecommons.org/licenses/by/4.0/, accessed on 11 January 2024).

**Table 2 ijms-25-01554-t002:** Genetic characteristics of thymoma-associated myasthenia gravis (TAMGs) and their interaction with signaling pathways.

Most Overexpressed Genes in TAMG	Most Downregulated Genes in TAMG	MiRNA Expression in TAMG	Upregulated Pathways in TAMG	Downregulated Pathways in TAMG
ATM [34], SFTPB [34], ANKRD55 [34], BTLA [34], CCR7 [34], TNFRSF25 [34], PNI SR [30], CCL25 [30], NBPF14 [30], PIK3IP1 [30], RTCA [30]	GADD45B [30], SERTAD1 [30], TNFSF12 [30], MYC [30], ADPRHL1 [30]	Reduced miR-20b expression [35]: function: anti-oncogene [35]; targets NFAT5 and CAMTA1 [35]	TGF-beta with the key component CCL25 [30], and HTLV-I pathway [30]	Hindered NFAT signaling by suppressing the expression of NFAT5 and CAMTA1 through reduced miR-20b expression [35]

## Data Availability

The data presented in this review are available on request from the corresponding author.

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
