# Peer review of "The Molecular Landscape of Thymic Epithelial Tumors: A Comprehensive Review"

_ijms, 2024, doi:10.3390/ijms25031554_

Round 1

Reviewer 1 Report

Comments and Suggestions for Authors

It is an interesting review on the thymus epithelial tumors topic, which can help researchers working on the molecular levels of thymus epithelial tumors. However, this review needs some minor modifications before it's acceptance. 

1. In the Abstract, the first 2 lines should always represent background of the topic.

2. Why authors didn't explain about ChIP-seq in section 2.4, before directly describing the research papers? I recommend to write few introductory lines on this topic.

3. Add more details with research papers for section 2.5.

4. Last paragraph of 2.6 section doesn't provide sufficient information, please add more info.

5. Section 2.7 heading, expand it. Please don't use abbreviations in headings and sub-headings.

6. Section 4.5 to 4.8 lacks sufficient details. Please discuss atleast 3-4 research papes here. Also, add few introductory lines before start describing the research papers. 

7. Graphical abstract is needed.

8. Throughout the manuscript, I couldn't see a single figure. I recommend to add atleast 3-4 figures inside the manuscript. Also, reproduce 3-4 figures from research papers by obtaining permission.

9. Section 5 needs to be accompanied with a small sub-section describing target therapy in TETs.

Comments on the Quality of English Language

Minor English language correction is required.

Author Response

Author's Reply to the Review Report

Thank you for your interest in our manuscript and for your valuable comments. The changes related to Reviewer 1 are marked in red in the text, and those related to Reviewer 2 are marked in green in our revised version of our manuscript.

Reviewer 1

  1. In the Abstract, the first 2 lines should always represent the background of the topic.

Thank you for your comment. The first two sentences now better present the general background of the topic.

  1. Some introductory lines to 2.4 CHIP-Seq section

Some introductory lines regarding 2.4 ChIP-Seq were added.

  1. Add more details with research papers for 2.5

Eight research papers with comprehensive results were added to section 2.5, as also suggested by Reviewer 2, along with a supplementary description of the methods.

  1. Last paragraph of 2.6 doesn't provide sufficient information, add more details

More Details to this section 2.6 were added.

  1. Section 2.7 heading, expand it. Please don't use abbreviations in headings and sub-headings.

We have now removed all abbreviations from the headings and sub-headings, including the heading of section 2.7.

  1. Section 4.5-4.8 lacks sufficient details. Please discuss at least 3-4 research papers here. Also, add few introductory lines before start describing the research papers.

Previous sections 4.2, 4.5 and 4.7 have been combined to a new section (Section 4.2 in the revised version of the manuscript) since RAS, MAPK etc. are genes of the EGFR and IGF-1R signaling pathway. Sections 4.3 and 4.4 have been also logically integrated (in a new section 4.3), as these pathways also interact with each other. Additional research papers in these new sections have been included. We also added some introductory lines before start describing the research papers.

Section 4.8 (now 4.4 in the revised version) has also been extended with two more additional research papers.

  1. Graphical abstract is needed.

A graphical abstract has been added.

  1. Add at least 3-4 figures inside the manuscript. Also, add at least 3-4 figures from research papers by obtaining permission.

We have now included 7 figures from published research papers and designed a new figure which illustrates comprehensively the mentioned pathways and their interaction.

  1. Section 5 needs to be accompanied with a small sub-section describing target therapy in TETs

We have inserted a subheading (5.1 Target Therapy in TETs) for target therapy into the chapter. Otherwise, the chapter on targeted therapy is extensively described.

-Minor English language correction is required.

A comprehensive review regarding the English language has been made.

Reviewer 2 Report

Comments and Suggestions for Authors

The paper authored by Lisa Elm and Georgia Levidou, titled "The Molecular Landscape of Thymus Epithelial Tumors: A Comprehensive Review," covers the thymic epithelial tumors and thymic carcinoma concerning their variable clinical presentations. Signal transduction pathway analysis is emphasized as crucial for understanding disturbances in the regulation of important cellular functions and pathways, as the authors state. Overall, the paper is well-written with valid references, but to truly achieve a 'comprehensive review' as implied by the title, substantial improvements are necessary.

Major points:

  1. 1. The paper lacks graphical figures throughout, which can make it tiresome for readers. Especially in the introduction part, a schematic diagram summarizing the authors' intentions along with detailed explanations is essential.

  2.  
  3. 2. On page 2, lines 59-75, the references for this section are inadequate. Considering the broad spectrum of content related to 'Next Generation Sequencing,' the number of references is minimal and even redundant. The richness of references is crucial for a review paper.

  4.  
  5. 3. Lines 98-105 on page 3 lack conciseness and intensity in presenting information. Adding overview tables throughout the paper for such sections would facilitate the extraction of important information.

  6.  
  7. 4. In section 2.4 to 2.5 on page 5, if these are considered key techniques by the authors, they should not be described so briefly. This paper is not merely a methodology description. The authors need to elaborate on how the described techniques relate to the theme and why they are essential.

  8.  
  9. 5. On page 9, in whole text 4, figures depicting pathways for quick comprehension are needed. Describing them solely in text feels uninspiring. This section needs significant improvement.

  10.  

Minor point: Care should be taken in describing abbreviations. Without listing them all, be sure to address this issue carefully.

Author Response

Author's Reply to the Review Report

Thank you for your interest in our manuscript and for your valuable comments. The changes related to Reviewer 1 are marked in red in the text, and those related to Reviewer 2 are marked in green in our revised version of our manuscript. 

Reviewer 2:

  1. Figures missing, especially in the introduction part a schematic diagram summarizing the author's intention along with detailed explanations is required.

Thank you for your comment. A Graphical Abstract has now been inserted. Additionally, as noted in our comments to Reviewer 1, further figures have been included in the revised version of our manuscript.

  1. On page 2, lines 59-75 , the references for this section is inadequate. Considering the broad spectrum of content related to NGS the number of references is minimal and redundant.

Lines 59-133 collectively pertain to NGS and already encapsulate many summarized contents from research papers of content related to NGS. Seventeen sources are cited in this section (see line 64 in the revised version of the manuscript). Lines 59-75 in the original version essentially summarize the main findings.

  1. Lines 98-105 in page 3 lack conciseness and intensity in presenting information. Adding overview tables throughout the paper for such section would facilitate the extraction of important information.

Table 1 provides an overview of genetic alterations among the several TETs, incorporating all these genes from lines 93-105 with corresponding source reference, in order to facilitate the extraction of the information presented in the text.

  1. In Section 2.4 and 2.5 on page 5, if these are considered key techniques by the authors, they should not be described so briefly. This paper is not merely a methodology description. The authors to elaborate on how the described techniques relate to theme and why they are essential.

The methodology description for 2.4 (also requested by Reviewer 1) has been accordingly supplemented. The same applies to section 2.5. An effort to explain why these techniques are important in understanding the molecular background of TETs has been made.

  1. On Page 9, in whole text 4, figures depiciting pathways for quick comprehension are needed. This section needs significant improvement.

In order to improve this section we inserted firstly two figures depicting the key signaling pathways in Part 4, pages 11-15. Secondly, we combined previous sections 4.2, 4.5 and 4.7 to one section (4.2 in the revised version) due to the fact that RAS, MAPK etc. are genes of the EGFR and IGF-1R signaling pathway. The same applies to sections 4.3 and 4.4, which have now been logically integrated (in a new section 4.3), as these pathways also interact with each other. Additional research papers in these new sections have been included, as also requested by Reviewer 1. Some introductory lines before describing the results of the research papers have also been added. The previous section 4.8 (now 4.4 in the revised version) has been supplemented with 2 more citations.

Additionally, several illustrative figures from the research papers were inserted to further illustrate the written text. A figure which illustrates comprehensively the mentioned pathways, and their interaction was also designed.

Minor point: Care should be taken in describing abbreviations. Without listing them all, be sure to address this issue carefully.

The abbreviations mentioned in the text have been rechecked for completeness.

Round 2

Reviewer 1 Report

Comments and Suggestions for Authors

The authors have addressed all of my queries.

Reviewer 2 Report

Comments and Suggestions for Authors

The authors have faithfully fulfilled my requirements.

Now, this paper is fully prepared for publication in IJMS.